# *Leuconostoc pseudomesenteroides* Bacteremia in an Immunocompromised Patient with Hematological Comorbidities—Case Report

**DOI:** 10.3390/microorganisms12112215

**Published:** 2024-10-31

**Authors:** Madalina Simoiu, Mihai-Cezar Filipescu, Meilin Omer, Alina Maria Borcan, Mihaela-Cristina Olariu

**Affiliations:** 1University of Medicine and Pharmacy “Carol Davila”, 020021 Bucharest, Romania; madalina.simoiu@umfcd.ro (M.S.); mihai-cezar.filipescu@rez.umfcd.ro (M.-C.F.); mihaela.olariu@umfcd.ro (M.-C.O.); 2National Institute for Infectious Diseases “Matei Bals”, 021105 Bucharest, Romania; 3Colentina Clinical Hospital, 020125 Bucharest, Romania; meilin_26@yahoo.com

**Keywords:** *Leuconostoc*, bacteremia, immunocompromised, leukemia

## Abstract

*Leuconostoc pseudomesenteroides* is a rare pathogen that can cause bacteremia in immunocompromised patients, particularly those with hematological conditions like acute myeloid leukemia. In this case, a 56-year-old woman developed *Leuconostoc* bacteremia following chemotherapy and multiple infections, including invasive aspergillosis. Despite broad-spectrum antibiotic treatments, her fever persisted until the blood cultures identified *Leuconostoc pseudomesenteroides*. Switching to intravenous ampicillin led to the resolution of symptoms. This case highlights the challenges in diagnosing and treating rare, glycopeptide-resistant bacteria in immunosuppressed patients and underscores the importance of repeated blood cultures and automated diagnostic systems. It also highlights the need for more rapid ways of diagnosing and treating such rare infections.

## 1. Introduction

*Leuconostoc* species, a member of the *Leuconostocaceae* family, is classified as a species with low pathogenicity. It is a type of bacteria that is Gram-positive, does not produce catalase, can survive with or without oxygen, produces lactic acid, and is commonly found in starter cultures used in the food industry as well as in nature on leafy vegetables or in wine [1,2,3]. They are not part of the human microbiota but can be found in some stool samples and the vagina [1]. When causing human infections, this species poses a problem, as it is naturally resistant to vancomycin and teicoplanin [4]. There are limited reported cases of *Leuconostoc* spp. bacteremia in the literature and a few articles addressing this species. In immunocompromised patients, *Leuconostoc* spp. typically causes infections as an opportunistic pathogen, as seen in this case, and leads to poor or even fatal outcomes [1,4].

The first case of *Leuconostoc* infection in humans was reported in 1985 by Buu-Hoi et al., but the prevalence and incidence of this organism in the hospital setting are unknown, primarily because of the misidentification with *Streptococcus* spp. or *Enterococcus* spp. [4].

Diagnosing this bacteria’s presence is challenging due to its uncommon occurrence. In the case presented, the patient had a positive blood culture, and a pure culture of *Leuconostoc* was obtained on Columbia blood agar. Gram staining revealed the presence of Gram-positive cocci in chains or pairs. The Vitek^®^ MS Prime (bioMérieux, St. Louis, MO, USA) automated system for microbial identification and antimicrobial susceptibility was used, resulting in *Leuconostoc*. In many laboratories where identification is limited to biochemical techniques, this bacterium can be easily confused with *Streptococcus* spp. or *Enterococcus* spp. [4]. In the absence of automated methods for microbial identification, there are few criteria for distinguishing *Leuconostoc* species: Gram-positive cocci, vancomycin resistance, a lack of catalase production, negative Voges–Proskauer test, production of gas from glucose in de Man Rogosa and Sharpe (MRS) agar, a lack of arginine deamination, and negative reactions to pyrrolidonyl arylamidase (PYR) [4].

## 2. Case Report

A 56-year-old woman, normal weight, with no prior health conditions, undergoing a routine evaluation for generalized physical weakness in a regional hospital, was diagnosed with thrombocytopenia and referred to a hematology unit in a large multi-disciplinary hospital in Bucharest. In March 2024, she was diagnosed with acute myeloid leukemia and was admitted to said hospital to undergo chemotherapy. The clinical presentation on the day of the admission is as follows: afebrile, intense pallor, petechiae on the lower limbs, tender, enlarged laterocervical and submandibular lymph nodes, white dehydrated tongue, white patches on both tonsils—with dysphagia and odynophagia, without clinical signs of organomegaly.

Upon admission, the blood work (Table 1) showed a marked increase in inflammation markers: CRP 190 mg/L (approximately 38 times the upper reference value), procalcitonin 0.23 ng/mL (approximately 4 times the upper reference value), and mildly elevated leukocytosis 10,930/mm^3^ with an increased ratio of lymphocytes and monocytes (lymphocytes 2900/mm^3^; monocytes 4920/mm^3^). An increased LDH value was noted—456 U/L (2 times the upper reference value). A blood smear showed anisocytosis thrombocytopenia. The myelogram revealed rich cellularity, approximately 85–90% monocytoid blast cells, frequent mitoses, a myeloid line with 15% megaloblasts, and megakaryocytes with non-platelet-specific granules. The overall aspect was suggestive of acute myeloid leukemia with 90% blast cells.

Immunophenotyping, molecular biology, and medullar aspiration established the diagnosis of acute myeloid leukemia—FLT3+. Given the clinical presentation and the high clinical suspicion for acute tonsillitis, an ENT (ear, nose, and throat specialist) consult was solicited, and a diagnosis of acute ulcerative tonsillitis was established. Pharyngeal microbial specimens were collected for culture analysis without isolating the etiological agent. Thus, empiric antibiotic therapy with IV ceftriaxone 2 g per day was started on the second day of hospitalization, with resolution of the symptoms for the next 10 days. Induction chemotherapy with cytarabine 180 mg per day, idarubicin 20 mg/day, and midostaurin (with dosage adjustment according to QTc values) was initiated in the first week. The treatment was well tolerated, and medullar aplasia was attained.

Approximately 2 weeks after admission, the patient developed a fever and local inflammation pertaining to the percutaneous central venous line catheter adjacent to the area of insertion. Blood cultures were collected without isolating a specimen. A diagnosis of phlebitis was established, and antimicrobial treatment was escalated to piperacillin-tazobactam (4.5 g q8h) and vancomycin (1 g q12h) for 12 days. The central venous line catheter was subsequently removed. Given the intense immunosuppressed state of the patient coupled with the continuous worsening of the general status and facing the impossibility of excluding an invasive fungal infection, an antifungal agent was added 2 days later—caspofungin (loading dose 70 mg/m^2^, then 50 mg/m^2^). Clinically, the evolution was favorable under this antimicrobial regimen.

In the fourth week after admission, the patient presented with an intense, productive cough. A computed tomography (CT) was performed with an image highly suggestive of invasive aspergillosis (Table 1), coupled with two consolidations pertaining to both upper lobes. Bronchoalveolar lavage with isolation of *Aspergillus* spp. was performed, and the antifungal treatment was switched to oral Voriconazole (400 mg q12h—loading dose, then 200 mg q12h) for an indeterminate period of time, depending on the lesion’s radiologic evolution. Antibiotic treatment was, once again, escalated to meropenem (1 g q8h) and IV linezolid (600 mg q12h) for the rest of the hospitalization (approximately 2 weeks).

Under this treatment, the outcome was favorable, and the patient was discharged from the hospital, with a hematological reevaluation pending for the next month (May 2024).

A control bone marrow puncture at the next admission was performed, and the results showed rich cellularity with 3% blast cells. The patient was again admitted in order to undergo consolidation chemotherapy. Soon after hospitalization, a CT scan was performed, which illustrated the radiological resolution of the two consolidations in the upper lobes but with the persistence of a lesion consistent with the radiological diagnosis of aspergilloma (Figure 1).

Consolidation chemotherapy was initiated, and medullar aplasia with severe pancytopenia was attained. About 2 days after, the patient developed a fever (38.1 °C). Yet again, the blood cultures failed to reveal a specific pathogen. Broad-spectrum antibiotic therapy with piperacillin-tazobactam and vancomycin was started, under which the fever persisted. Blood cultures were again drawn from the patient, both from the central venous line catheter (a device that was removed and sent for microbiological testing; cultures came back negative) and from a peripheral vein. This time, after 5 days of incubation (possibly indicative of a somewhat low bacterial concentration) in the BD BACTEC^™^ FX40 system, both sets of blood cultures revealed bacterial growth. The blood cultures were inoculated on a solid blood agar medium for 24 h at a temperature of 37 °C, with a sufficient bacterial yield. Afterward, an automated system (Vitek^®^ MS Prime) was used, using the Vitek^®^ protocol (0.5 McFarland units inoculated in the Vitek Gram-positive card). Using data from the medical literature, antimicrobial therapy was, therefore, changed to IV ampicillin (2 g q4h) for a total of 14 days, with a resolution of the fever. Blood cultures were repeated in order to assess the efficacy of the treatment, with no bacterial growth. The patient was discharged after the abatement of the medullar aplasia but with the persistence of moderate anemia (Figure 2).

## 3. Discussions

The case presented is suggestive of the opportunistic nature of *Leuconostoc* species. This is a very rare cause of bacteremia linked with nosocomial [5] infections. These types of infections often have a negative, sometimes even fatal, outcome. The sheer complexity of the case presented the intricate nature of the infections (including here invasive aspergillosis) and shed new light on the capabilities of modern medicine. Often confused with other Gram-positive species (*Streptococcus* spp., *Enterococcus* spp.) and with an innate resistance to glycopeptides, this etiology poses significant difficulties, especially from a diagnostic point of view. Although rarely described in clinical practice, *Leuconostoc pseudomesenteroides* infections are most certainly underdiagnosed, given the difficulties posed by microbiological testing. Given the fact that many empiric antimicrobial regimes used for immunocompromised patients contain Vancomycin or Teicoplanin, the persistence of fever in the setting of maximal antimicrobial treatment might raise suspicion for alternative etiologies, including *Leuconostoc pseudomesenteroides*.

Needless to say, the case presented proves the fact that automated systems for microbial detection and antimicrobial susceptibility testing do enhance the diagnostic and treatment options available. In resource-limited settings, where such systems are not available yet, such rare infections might not be diagnosed in time, with dire consequences for the patient involved.

Even if the outcome was favorable in our case, this case report has limitations. One of them is the fact that we did not have a profile of antimicrobial susceptibility for the *Leuconostoc pseudomesenteroides* isolated from the blood cultures. The laboratory did not possess the necessary reference values (according to CLSI 2024) in order to interpret said antibiogram. Another limitation is the fact that this infection cannot be linked with a specific syndrome pertaining to a specific organ, given the fact that the central venous line catheter cultures did not yield the same result as the two successive blood cultures.

Clinicians should also focus on prevention rather than diagnosis and/or treatment. Thoroughly respecting the existing guidelines for disinfection and antiseptic practices might prevent such critical situations from developing. Also, the importance of antibiotic stewardship can never be understated.

It is important to bear in mind that even state-of-the-art automated systems have their limitations, as they are dependent on positive blood cultures. The importance of more traditional ways of bacterial identification cannot be overstated.

It is also cardinal to note the importance of drawing repeated blood cultures from a patient presenting with a fever unresponsive to maximal empiric antibiotic therapy. More blood cultures enhance the sensibility of this very important tool in our diagnostic arsenal. Unfortunately, there are certain situations in which repeated blood cultures are simply not possible to draw, such as a spent venous capital or in a neonatology section (where the blood volume is already limited). In such cases a more personalized approach should be taken.

## 4. Conclusions

*Leuconostoc pseudomesenteroides* is a Gram-positive bacterium used in food fermentation, rarely known to cause infections in humans. This facultative anaerobic, catalase-negative, intrinsically resistant to glycopeptides is known to cause infections in immunocompromised patients, confirming its opportunistic and nosocomial potential. The case presented, although somewhat reminiscent of other cases described in medical literature [6], was not without the inherent complexities of acute myeloid leukemia. From the repeated tonsillitis with an unknown pathogen to the invasive lung aspergillosis and subsequently, to the diagnosis of bacteremia with *Leuconostoc pseudomesenteroides*, this case posed a significant number of infectious complications.

Repeated blood cultures [7] and automated systems [8], such as the one used in our case, are important tools in order to enhance the diagnosis and treatment of opportunistic and hospital-acquired infections. The case presented had a favorable outcome only after initiating high-dose ampicillin. It is important for medical practitioners all around the world to keep this albeit rare agent in mind, especially in the setting of a refractory fever under maximal empiric antimicrobial therapy (including here, anti-MRSA agents, such as glycopeptides).

## Figures and Tables

**Figure 1 microorganisms-12-02215-f001:**
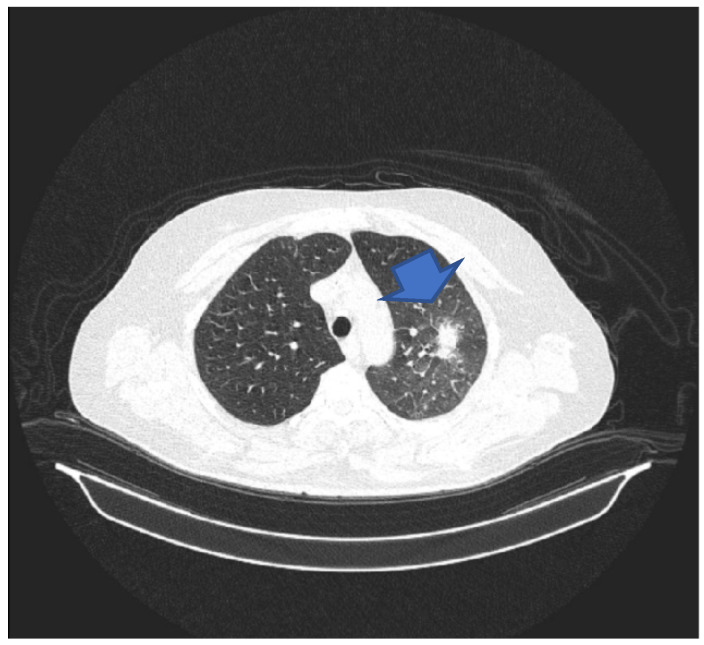
CT scan suggestive of pulmonary invasive aspergillosis. The arrow shows the location of the lesion.

**Figure 2 microorganisms-12-02215-f002:**
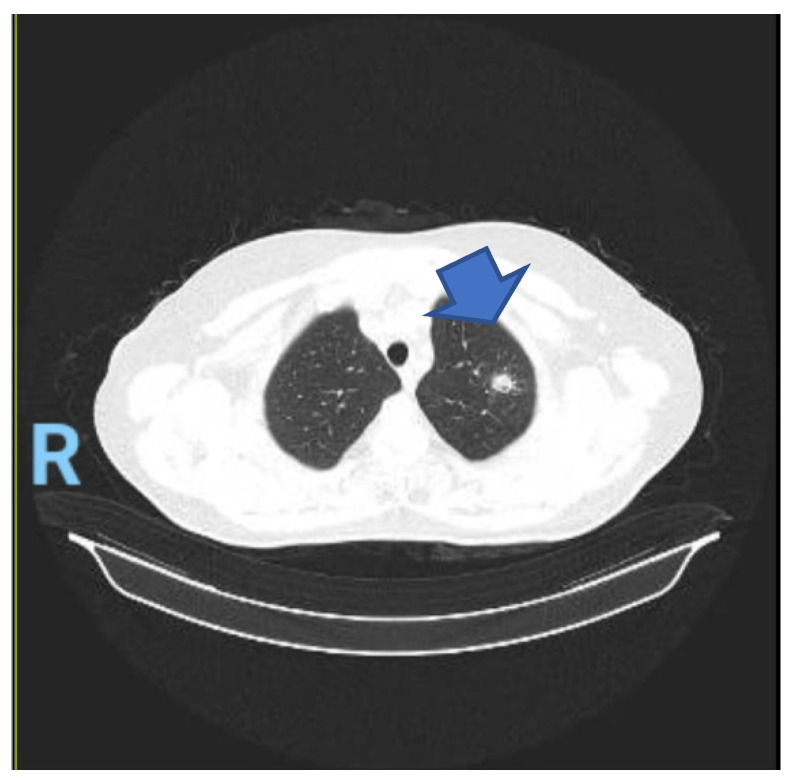
Reevaluation CT scan after the initiation of Voriconazole for invasive aspergillosis. The arrow indicates the location of the lesion. The letter “R” is used for spatial orientation by the computed tomography’s software.

**Table 1 microorganisms-12-02215-t001:** Table highlighting laboratory parameters upon admission.

Parameter	Value	Reference Value
CRP	190 mg/L	<5 mg/L
Procalcitonin	0.23 ng/mL	<0.5 ng/mL
WBC	10,930/mm^3^	4000–10,500/mm^3^
Neutrophils	3500/mm^3^	2000–8000/mm^3^
Lymphocytes	2900/mm^3^	1000–4000/mm^3^
Monocytes	4290/mm^3^	200–1000/mm^3^
LDH	456 U/L	135–214 U/L

## Data Availability

Data are available upon request from the authors.

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
