# Peer review of "Leuconostoc pseudomesenteroides Bacteremia in an Immunocompromised Patient with Hematological Comorbidities—Case Report"

_microorganisms, 2024, doi:10.3390/microorganisms12112215_

Round 1
Reviewer 1 Report
Comments and Suggestions for Authors
The author summarizes in this case report, highlighted the challenges in diagnosis and treatment rare, glycopeptide-resistant bacteria in immunosuppressed patients”. The report also describes the importance of repeated blood cultures and automated diagnostic systems. The report is overall well written and under-stable for readers.
Author Response
We, the authors, have taken into consideration the reviewers’ comments and have made the necessary modifications to make the article more reader-friendly, with a better scientific narrative and clearer, more concise conclusions. We wish to thank the reviewers for their time and constructive criticism. It has certainly helped us create a better scientific paper. Not in a particular order we have:
- Correctly written the names of antibiotics;
- Correctly italicized bacterial names, for a better identification;
- Specified which type of Vitek automated system was used;
- We do not consider using a table for the lab parameters would enhance the reading experience, given the fact that there are not many such parameters presented;
- Specified which type of agar was used;
- Highlighted the importance of antimicrobial stewardship and prevention techniques in such cases of rare and life-threatening infections, especially in the setting of immunodepression;
- Modified the abstract in order to better convey the message of the article;
- Specified which type of catheter was used and the fact that it was removed;
- Unfortunately we cannot possibly replace the images used for the CT scans. It is our opinion that they are highly suggestive of invasive aspergillosis; also added arrows for a better identification of the lesions;
- Specified the duration of ampicillin therapy and the results of subsequent blood cultures;
- We specified in the article the absence of guidelines (EUCAST or CLSI) regarding Leuconostoc pseudomesenteroides and the fact that we did not have a proper antibiogram We listed this fact as a limitation of the study. It also highlights the need for a better cooperation with the microbiology laboratory.
- We specified the reference values for the modified biological parameters
- Unfortunately, we do not have an image or a photograph of the plate used for bacterial culture
We wish to thank the reviewers for their time and guidance regarding this article.
Reviewer 2 Report
Comments and Suggestions for Authors
Summary of the Study:
This study presents a compelling case of a 56-year-old immunocompromised patient with acute myeloid leukemia who suffered from multiple secondary infections, notably aspergillosis. After extensive testing and treatment, the source of persistent fever was ultimately identified through blood cultures as a rare bacteremia caused by Leuconostoc pseudomesenteroides. The study emphasizes the challenges of diagnosing rare bacteremia, particularly in hospital-acquired infections, and advocates for the use of automated systems like Vitek-bioMérieux to improve diagnostic accuracy and treatment outcomes.
Major Comments:
- Please specify which Vitek system was utilized in the study, such as VITEK® 2 XL, VITEK® 2 Compact 60, or VITEK Reveal, in lines 46 and 47.
- To enhance understanding for a diverse readership, please consider adding arrows in Figures 1 and 2 to highlight the regions affected by aspergillosis.
- Please include a brief overview of the sample preparation steps required for analysis using the Vitek automated system.
- For clarity, it would be helpful to present all the lab parameters mentioned in lines 65 to 72 in a table format, along with their reference ranges. Please include a table
- While automated systems such as Vitek have advantages, they depend on positive blood cultures. If bacterial colonies do not form, the automated systems may not enhance diagnosis as identification and antibiotic therapy are only possible after a couple of days (starting from positive blood cultures to identification and AST determination in a few hours). This emphasizes the importance of effective plating techniques, which still take 2 to 4 days based on bacterial growth kinetics. How can this limitation be addressed, especially for slow-growing and rare bacterial types?
- The Vitek system has a specific bacterial panel, and it was fortunate that this case involved a bacterium within that panel. Please discuss how to choose the best automated system given that no system can detect all possible bacteria, and consider the implications of detection range, sensitivity, and specificity.
- Drawing multiple blood samples at different times can be difficult, especially for neonates, where volume is limited. What strategies can be implemented to address this challenge?
- The typical growth time for Leuconostoc pseudomesenteroides on agar is 1 to 3 days. In this case, the organism was detected after 5 days of incubation following initial negative cultures. Could this indicate a low bacterial concentration, or was there an issue with the type of agar used? Please specify the agar type utilized for blood cultures in this study.
- Given these complexities, it may be more beneficial to invest in prevention strategies rather than solely focusing on identifying organisms and developing antibiotic therapies. Please discuss the importance of antibiotic Stewardship, sterilization/disinfection, education, and training in the discussion section to address the prevention of secondary infections in immunocompromised patients such as cancer patients.
Author Response

(The authors gave the same response as above.)

Reviewer 3 Report
Comments and Suggestions for Authors
Thank you for inviting me to review this manuscript. It is interesting and well-written. I have some comments that could be of use:
· Line 23: no need for ampicillin to start with a capital a (the same is for other antibiotics and the chemotherapeutic drugs later on)
· The conclusion of the abstract should underline the need for rapid and accurate pathogen identification methods that would allow adequate diagnosis and management of these infections
· Line 33: I think that the reference style for the journal is different
· Line 41: all microorganism names should be in italics throughout the manuscript
· Line 43-54: This part should not be in the introduction section. Details of the case belong in the case report, and the last part belongs to the discussion section
· Line 66: It would be a good idea to mention the values and the reference values throughout the case report
· Line 84: pertaining to the CVC à how can you be so sure? Did you remove the catheter?
· Line 92: It is better not to start a sentence with a number. If it is inevitable, write it in full
· Figure 2 is of low quality. Could it be replaced? The ‘R’ sign is not needed
· Line 123: It is not clear if the CVC was removed. Was it a temporal catheter, a PICC, or a Port?
· For how long was ampicillin given? Was a repeat blood culture taken?
· How was the identification performed? How was the susceptibility testing performed, and based on what method of interpretation? Was it EUCAST or CLSI?
· Antimicrobial susceptibility should be provided in a table
· Ideally, a figure of the plate where the colonies were cultured should be also shown
· English should be improved
· The manuscript requires a lot of improvement before it can be considered of publication quality
Comments on the Quality of English LanguageModerate changes needed
Author Response

(The authors gave the same response as above.)

Round 2
Reviewer 2 Report
Comments and Suggestions for Authors
Unfortunately, the authors have not adequately addressed the questions raised. Key questions such as 3, 5, 6, and 7 have been completely overlooked, with no discussion or explanation provided. Furthermore, one of the suggestions was to include an arrow in Figures 1 and 2. While the authors did add an arrow, its poor aesthetic significantly detracts from the figures' overall appeal.
Additionally, the authors failed to indicate the line numbers where changes were made, which is unacceptable. Reviewers aim to help improve the manuscript, but the minimal effort put forth by the authors in revising the paper suggests otherwise. Many responses were insufficient, and this manuscript cannot be accepted until all reviewer questions are fully addressed and the concerns raised are adequately discussed.
Author Response
Response to reviewer 2.
Thank you very much for the time and effort that you have dedicated to provide us your valuable suggestions for improving our manuscript “Leuconostoc pseudomesenteroides bacteremia in an immunocompromised patient with hematological comorbidities – Case Report”. We will give adequate, this time, responses to each and every problem with our manuscript, as highlighted by the reviewer.
Comment 1: Please specify which Vitek system was utilized in the study, such as VITEK® 2 XL, VITEK® 2 Compact 60, or VITEK Reveal, in lines 46 and 47.
Response: Vitek® MS was used. See line 48.
Comment 2: To enhance understanding for a diverse readership, please consider adding arrows in Figures 1 and 2 to highlight the regions affected by aspergillosis.
Response: We have changed the arrow style to ensure a better highlighting of the pulmonary lesion suggestive of invasive aspergillosis.
Comment 3: Please include a brief overview of the sample preparation steps required for analysis using the Vitek automated system.
Response: A brief overview of the sample preparation was included in the reviewed draft. Unfortunately we forgot to specify in the cover letter said modification. See lines 127-133.
Comment 4: For clarity, it would be helpful to present all the lab parameters mentioned in lines 65 to 72 in a table format, along with their reference ranges. Please include a table.
Response: We have included a table which shows laboratory values upon admission and their respective reference values. See line 77.
Comment 5: While automated systems such as Vitek have advantages, they depend on positive blood cultures. If bacterial colonies do not form, the automated systems may not enhance diagnosis as identification and antibiotic therapy are only possible after a couple of days (starting from positive blood cultures to identification and AST determination in a few hours). This emphasizes the importance of effective plating techniques, which still take 2 to 4 days based on bacterial growth kinetics. How can this limitation be addressed, especially for slow-growing and rare bacterial types?
Response: As the reviewer correctly pointed out, the importance of effective and targeted plating techniques cannot be overstated. This case report’s aim is to highlight the possibility of infection with glycopeptide-resistant Gram-positive bacteria, in an immunosuppressed setting. It is our opinion that the focus must be put on prevention. Even though there are specific cases (such as this one) in which prevention is impossible, more effort should be put into limiting colonization with resistant bacteria, especially considering the fact that patients with hematological comorbidities are often subjected to long stays in the hospital. Our research took into account the limitations of automated systems, such as the dependence on positive blood cultures. Highlighting the importance of more traditional ways of bacterial identification is paramount to a comprehensive and complete diagnosis. See Lines 168-170.
Comment 6: The Vitek system has a specific bacterial panel, and it was fortunate that this case involved a bacterium within that panel. Please discuss how to choose the best automated system given that no system can detect all possible bacteria, and consider the implications of detection range, sensitivity, and specificity.
Response: A “best” automated system should be chosen based on resource availability, epidemiological trends and particularities pertaining to every hospital or unit. In certain scenarios, other automated systems could have been used, such as MALDI-TOF Bruker ®, which are definitely way more dependable, given their lower detection time, their higher specificity and sensibility but also more expensive. It was fortunate that in our case the automated system, Vitek, had the panel for Gram-positive bacteria, including Leuconostoc pseudomesenteroides.
Comment 7: Drawing multiple blood samples at different times can be difficult, especially for neonates, where volume is limited. What strategies can be implemented to address this challenge?
Response: We have added a paragraph answering this question, lines 174-177. We hinted towards a more tailored approach to each patient.
Comment 8: The typical growth time for Leuconostoc pseudomesenteroides on agar is 1 to 3 days. In this case, the organism was detected after 5 days of incubation following initial negative cultures. Could this indicate a low bacterial concentration, or was there an issue with the type of agar used? Please specify the agar type utilized for blood cultures in this study.
Response: We have indicated the possibility of a low bacterial concentration in the manuscript, line 128.
Comment 9: Given these complexities, it may be more beneficial to invest in prevention strategies rather than solely focusing on identifying organisms and developing antibiotic therapies. Please discuss the importance of antibiotic Stewardship, sterilization/disinfection, education, and training in the discussion section to address the prevention of secondary infections in immunocompromised patients such as cancer patients.
Response: We have discussed the importance of prevention, in conjunction with diagnosing and treating such cases. Lines 164-168.

Reviewer 3 Report
Comments and Suggestions for Authors
The manuscript has been improved.
Comments on the Quality of English LanguageMinor
Author Response
We, the authors, thank you for your time and your feedback. Your input was definitely necessary in order to ensure a better, more qualitative article.